# Genetic Diversity and Population Structure of Chinese Chestnut (*Castanea mollissima* Blume) Cultivars Revealed by GBS Resequencing

**DOI:** 10.3390/plants11243524

**Published:** 2022-12-14

**Authors:** Xibing Jiang, Zhou Fang, Junsheng Lai, Qiang Wu, Jian Wu, Bangchu Gong, Yanpeng Wang

**Affiliations:** 1Research Institute of Subtropical Forestry, Chinese Academy of Forestry, Hangzhou 311400, China; 2State Key Laboratory of Tree Genetics and Breeding, Chinese Academy of Forestry, Beijing 100091, China; 3Qingyuan Bureau of Natural Resources and Planning, Lishui 323800, China

**Keywords:** Chinese chestnut, SNP-based genetic diversity, population structure, geographical division

## Abstract

Chinese chestnut (*Castanea mollissima* Bl.) is one of the earliest domesticated and cultivated fruit trees, and it is widely distributed in China. Because of the high quality of its nuts and its high resistance to abiotic and biotic stresses, Chinese chestnut could be used to improve edible chestnut varieties worldwide. However, the unclear domestication history and highly complex genetic background of Chinese chestnut have prevented the efficiency of breeding efforts. To explore the genetic diversity and structure of Chinese chestnut populations and generate new insights that could aid chestnut breeding, heterozygosity statistics, molecular variance analysis, ADMIXTURE analysis, principal component analysis, and phylogenetic analysis were conducted to analyze single nucleotide polymorphism data from 185 Chinese chestnut landraces from five geographical regions in China via genotyping by sequencing. Results showed that the genetic diversity level of the five populations from different regions was relatively high, with an observed heterozygosity of 0.2796–0.3427. The genetic diversity level of the population in the mid-western regions was the highest, while the population north of the Yellow River was the lowest. Molecular variance analysis showed that the variation among different populations was only 2.07%, while the intra-group variation reached 97.93%. The Chinese chestnut samples could be divided into two groups: a northern and southern population, separated by the Yellow River; however, some samples from the southern population were genetically closer to samples from the northern population. We speculate that this might be related to the migration of humans during the Han dynasty due to the frequent wars that took place during this period, which might have led to the introduction of chestnut to southern regions. Some samples from Shandong Province and Beijing City were outliers that did not cluster with their respective groups, and this might be caused by the special geographical, political, and economic significance of these two regions. The findings of our study showed the complex genetic relationships among Chinese chestnut landraces and the high genetic diversity of these resources.

## 1. Introduction

*Castanea mollissima* Blume, which is commonly referred to as the Chinese chestnut, is a member of the family Fagaceae. Seven species of *Castanea* are widely distributed in the temperate zone of the northern hemisphere. Four species are distributed in Asia: *C. mollissima* Blume, *C. seguinii* Dode, and *C. henryi* (Skan) Rehd. et Wils. in China and *C. crenata* Sieb. et Zucc. (Japanese chestnut) in Japan and the Korean Peninsula. Two species are distributed in North America, *C. dentata* Borkh. (American chestnut) and *C. pumila* Mill. (American chinquapin), and one species, *C. sativa* Mill. (European or sweet chestnut), is distributed in Europe. Chinese chestnut, European chestnut, and Japanese chestnut are the main cultivated species, and they are widely planted because of their high yields of edible chestnuts and economic value. These species also have abundant germplasm resources, can grow under diverse environmental conditions, and bear nutritious fruits; Chinese chestnut is also an important source of genes for the improvement of edible varieties [1,2]. Asian chestnuts are generally much more resistant to biotic and abiotic stresses, especially Chinese chestnut. The production of chestnuts in Europe has declined gradually due to chestnut blight and ink disease since the 20th century, and American chestnut has nearly disappeared because of chestnut blight [3]. Given that Chinese chestnut shows high resistance to both diseases, it has been introduced to other regions to breed disease-resistant varieties [4]. Chinese chestnut thus plays an important role in ensuring the sustainable utilization of chestnut resources worldwide [5].

The origin of chestnut species and the center of genetic diversity of chestnuts is in mainland China [6]. The secondary origin and center of genetic diversity of chestnuts is thought to be in Turkey [7]. Chinese chestnut, one of the earliest cultivated fruit trees in China, was first depicted in the “Book of Songs” and has been cultivated for at least 3000 years [8]. Chinese chestnut is widely distributed in China, especially in the Dabie Mountains and Yanshan Mountains, and it is cultivated in at least 22 provinces. The genetic structure and domestication history of Chinese chestnut remain unclear.

The development of next-generation sequencing technology has greatly increased sequencing throughput and reduced sequencing costs [9]. The genomes of an increasing number of plant species have been sequenced, and these new data have enhanced our understanding of the genomic attributes of plant populations.

The factors affecting variation in the genome and genetic variation among populations, such as natural selection, mutation, gene drift, and gene flow, can be inferred through the study of polymorphic sites based on high-coverage whole-genome data [10]. Population genomics studies of plants have been a major focus of research in the life sciences in recent years, and such studies have greatly enhanced our understanding of the roles of genetic recombination, linkage disequilibrium, and selection on the genomes of target populations. Population genomics technology has also contributed to our ability to explore population genetic structure, the origin of cultivated populations, and the molecular mechanisms underlying complex traits [7,8,9,10,11]. The distribution of nucleotide diversity also provides a useful tool for inferring population history and genetic diversity [12]. This has led to an increase in the number of studies examining non-model plants with high economic value [13].

Single nucleotide polymorphisms (SNPs) are single-base differences in the genome of different individuals of a species, and they are considered the most common type of genetic variation [14,15]. SNPs are third-generation molecular makers with several advantages: they are dimorphic, high density, genetically stable, easy to detect, and present in all DNA sequences in the genome [16,17]. With the development of high-throughput technology and bioinformatics, large-scale automated detection methods have been developed, and SNPs as an important marker was used for genetic map construction, biodiversity detection, and association analysis of linkage disequilibrium [18]. However, most genetic studies of chestnuts have been conducted using simple sequence repeats (SSRs) and other second-generation molecular markers; by contrast, few genetic studies of Chinese chestnut based on SNPs have been conducted, while more recent research utilized SNPs in European chestnut [10,19,20].

Exploration of the origin and evolution of plants is important for enhancing the development and utilization of plant resources and genetically improving existing cultivars. The traditional classification of chestnut varieties was based on their morphological characteristics (e.g., leaf, fruit, and stem), biological characteristics, and economic traits [21]. However, accurately identifying chestnut varieties based on the traditional classification method of using morphological or biological characteristics is a major challenge due to the long seed-bearing period, which is not beneficial to the production and improvement of woody plant varieties.

Genotyping by sequencing (GBS) is a particularly promising genomic approach that has proven to be an effective tool for genetic studies of plants, such as barley, switchgrass, yellow mustard, olive, and Norway spruce [22,23,24]. GBS technology was used to conduct genome-wide association studies and characterize patterns of genomic diversity. Few studies have used GBS technology to study Chinese chestnuts to date. GBS has been used to construct a high-density genetic map and identify QTLs related to the size and ripening period of Chinese chestnut fruit [25], but few studies of the genetic structure of Chinese chestnut populations have been conducted using GBS technology. With the release of the whole genome of Chinese chestnut, GBS technology could provide new insights into the origin and evolution of Chinese chestnut, and this could aid future efforts to genetically improve chestnut varieties.

Here, GBS was performed on 185 Chinese chestnut landraces from five regions (North of the Yellow River region, Eastern Coastal region, Yangtze River Basin region, South Central region, and Midwest region) to obtain genome-wide SNP data used to characterize genetic diversity and population structure of Chinese chestnut. ADMIXTURE analysis, principal component analysis (PCA), and phylogenetic analysis were performed to clarify the genetic relationships among Chinese chestnut landraces from different regions. We also aimed to clarify the role that humans have played in the diffusion of Chinese chestnut populations.

## 2. Material and Methods

### 2.1. Plant Material

A total of 185 different landraces of Chinese chestnut (*C. mollissima*) from five regions were collected, including 39 samples from the North of the Yellow River region, 90 samples from the Eastern Coastal region, 22 samples from the Yangtze River Basin region, 24 samples from the South Central region, and 10 samples from the Midwest region. These regions span 14 provinces in China, including Beijing City, Hebei Province, Anhui Province, Jiangsu Province, Shandong Province, Zhejiang Province, Shanghai City, Hubei Province, Hunan Province, Guangxi Province, Guangdong Province, Yunnan Province, Henan Province, and Shaanxi Province (Appendix A). With the exception of the hybrid clone of RISF-72 (*C. mollissima ‘Hongli’* × *Wild Chinese chestnut*), all other accessions are locally grown *C. mollissima* cultivars.

### 2.2. Sequencing and Mapping

Leaf tissues of Chinese chestnut samples were used for DNA extraction. Young chestnut leaves were harvested, immediately frozen in liquid nitrogen, and then transferred to −80 °C. The DNA was extracted according to Jiang et al., 2017 [11]. The GBS library was constructed via the following steps: (1) the genomic DNA was digested using restriction enzymes according to Elshire 2011 [26], and each sample was amplified and mixed after ligating the barcoded adapters; (2) required fragments were selected for library construction, and paired-end 150 sequencing was conducted using the Illumina HiSeq sequencing platform; and (3) Qubit 2.0 fluorometer (Invitrogen, Waltham, MA, USA) was used to preliminarily quantify the concentration of DNA after library construction, and the library was diluted to 1 ng/µL. Agilent 2100 Bioanalyzer (Agilent, Santa Clara, CA, USA) was used to detect the insert size of the library. When the insert size met expectations, the effective concentration of the library was quantified using qPCR (effective library concentration > 2 nM) to ensure the quality of the library. The different libraries were then pooled according to the effective concentration and target offline data volume and sequenced using Illumina Hiseq PE150 platform (Invitrogen, USA).

The raw data obtained from sequencing were filtered by quality control using the following criteria: (1) reads containing adapters were filtered; (2) when the N content (undetected base) in single-end sequencing reads exceeded 10% of the length of the read, the paired reads were removed; and (3) when the number of low-quality (≤5) bases contained in single-end sequencing reads exceeded 50% of the length of the read, the paired reads were removed. The products of the GBS digestion were statistically analyzed to evaluate enzymatic digestion efficiency. The number of reads with two ends generated by MseI and the ratio of the number of captured reads to the number of high-quality reads (enzymeCatchRatio) were determined. After obtaining the high-quality sequencing data, they were mapped to the reference genome (http://gigadb.org/dataset/view/id/100643, accessed on 15 August 2019) using the Burrows-Wheeler Alignment tool (BWA) [27]. The parameters (mem -t 4 -k 32 –M) were used; that is, local alignment was performed using four threads, the minimum seed length was 32, and shorter split hits were marked as secondary.

### 2.3. SNP Detection and Annotation

SAMTOOLS was used to detect population data in our study [28]. A Bayesian model was used to detect polymorphic loci in populations. To obtain high-quality data, SNPs with a read depth (dp) < 2, missing rate (Miss) > 0.2, and minor allele frequency (MAF) < 0.01 were removed. The obtained high-quality data were annotated using ANNOVAR software [29].

### 2.4. Population Stratification Analysis

Population genetic structure refers to the non-random distribution of genetic variation in a species or population. The subpopulations based on their geographical distributions or other criteria are usually geographically isolated individuals or populations. Different individuals within the same subpopulations are closely related to each other, and individuals of different subpopulations are more distantly related. ADMIXTURE is a program for the maximum likelihood estimation of individual ancestries based on large SNP genotype datasets. Here, it was utilized to analyze the population structure of Chinese chestnut. After generating the PLINK input Ped file, ADMIXTURE analysis was performed to characterize population genetic structure and identify population lineages.

PCA is a dimensionality reduction method that can transform the initial data set into a set of linearly uncorrelated variables. It depends on the date set and divides into two or three major axes of variation for visualization. PCA can also be used for cluster analysis. Eigenvectors and eigenvalues were calculated using GCTA software, and 2D and 3D PCA graphs were plotted using R. Individuals were clustered into different subgroups based on principal components according to SNP differences in individual genomes. We used this approach to explore the genetic structure of Chinese chestnut subpopulations.

TreeBeST software (1.9.2_i386, SourceForge Headquarters, San Diego, CA, USA) (http://treesoft.sourceforge.net/treebest.shtml, accessed on 19 October 2007) was used to generate the distance matrix. The neighbor-joining method was used to construct the phylogenetic tree of Chinese chestnut with 1000 bootstrap replicates.

## 3. Results

### 3.1. Sequencing Data Analysis

A total of 185 Chinese chestnut samples were used for sequencing analysis. The total sequencing data volume was 76.24665 Gb, with an average of 412.144 Mb per sample. Results showed that the sequencing quality was high (Q20 ≥ 88.27%, Q30 ≥ 81.43%) with a normal GC distribution (Table 1).

No samples were contaminated by adapter sequences, indicating a successful library construction. Then, sequencing data of 185 chestnuts were mapped to the reference genome (724,001,627 bp). The average mapping rate of the population samples was 91–98.13%, and the average sequencing depth of the genome was 6.51–14.75× with a 1× coverage rate (at least one base coverage) for over 5.08% of the samples (Table 2).

Next, we detected a total of 3,962,053 SNP sites using SAMTOOLS, which were filtered to obtain high-quality SNPs using the criteria of dp2, Miss0.2, Maf0.01, and the filtering criteria were performed according to the description of Lipka et al., 2012 [30]. A total of 299,015 SNPs were obtained for subsequent analysis (Table 3).

### 3.2. Genetic Diversity Analysis of Population

We found that the five populations have high heterozygosity with obvious differences (Table 4). Results showed that the values of observed heterozygosity and expected heterozygosity of the Midwest population were the highest, reflecting the highest level of genetic diversity. Furthermore, the South Central population’s genetic diversity level is close to that of the population of the Yangtze River Basin and the Eastern Coastal populations. However, the genetic diversity level of the population in the North of the Yellow River region was the lowest. Results of gene flow demonstrated higher chestnut genetic diversity in the Yangtze River Basin region, and relatively lower diversity in the Midwest region (Table 4).

Molecular variance analysis is an analysis of molecular differences. It defines different genetic structures and carries out statistical tests by classifying and dividing the studied populations at different levels, so as to estimate the proportion of differences within populations and among individuals in the total variation among populations. The variance component among the five Chinese chestnut populations was 8.759, and the variation percentage was 2.07%. Moreover, the variance component within the population was 414.740, and the variation percentage was 97.93%, which indicates that the genetic variation of the Chinese chestnut population mainly came from within the five populations (Table 5). These results were consistent with the results of phenotypic genetic variation of Chinese chestnut populations in our previous studies.

### 3.3. ADMIXTURE Analysis of 185 Chinese Chestnut Landraces

ADMIXTURE analysis was performed on 299,015 SNPs to explore the population structure of all Chinese chestnuts. The number of populations (K) was set from 1 to 12, and the K optimum was selected based on the cross-validation error compared to other K values (Appendix A). As we were focused on the regional distribution of chestnuts, especially the north–south division, we focused on cases in which k ranged from 2 to 5.

A total of 185 chestnuts were divided into two groups when k = 2 (Figure 1a, Appendix A). The first group had 67 samples, and the second group had 118 samples. A total of 28 samples in the first group and 11 samples in the second group were from the North of the Yellow River region. A total of 27 samples and 63 samples in the first and second group, respectively, were from the Eastern Coastal region. There were 5 and 17 samples in the first and second group, respectively, from the Yangtze River Basin region. There were 4 and 20 samples in the first and second group, respectively, from the South Central region. There were 3 and 7 samples in the first and second group, respectively, from the Midwest region. Samples from the North of the Yellow River region were mainly in the first group, and samples from the other regions were mainly in the second group. These distributional patterns are consistent with the north–south geographic division around the Yellow River. However, we also observed a few irregular samples, which were mainly from Beijing (North of the Yellow River region) and Shandong Province (Eastern Coastal region). Some gene flow appears to have occurred between these two groups.

All chestnut samples were divided into three groups of 98, 63, and 24 samples when k = 3 (Figure 1b). There were 10 samples in the first group, 28 samples in the second group, and 1 sample in the third group from the North of the Yellow River region. When k = 4 (Figure 1c), we obtained four groups of 91, 24, 40, and 30 chestnut samples. There were 9, 1, 21, and 8 samples in the first, second, third, and fourth groups, respectively, from the North of the Yellow River region. When k = 5 (Figure 1d), there were five groups of 10, 71, 34, 15, and 55 chestnut samples. There were 0, 12, 1, 1, and 25 samples in the first, second, third, fourth, and fifth groups, respectively, from the North of the Yellow River region.

### 3.4. PCA of 185 Chinese Chestnut Landraces

Tassel v5.0 software (https://tassel.bitbucket.io/, accessed on 23 February 2021) was used to analyze the results of PCA based on N × SNP matrix [28]. PCA was performed on 299,015 SNPs from all chestnut samples; they were annotated based on the results of the ADMIXTURE when k = 2. PC1, PC2, and PC3 explained approximately 15.7%, 8.1%, and 6.0% of the total variance, which captured ~1/3 of the genetic information of the samples. Pop1 was located on the left side of the PC1 axis, and pop2 was located on the right side of this axis, which indicates that PC1 clearly separated the two populations. However, these two populations were not separated along PC2 (Figure 2a). Most samples were clustered along PC3 in the PC2 vs. PC3 plot. Only a few pop1 samples were clearly separated from most samples along PC3, and a few pop2 samples were clearly separated from most samples along PC2 (Figure 2b). The two populations were clearly separated along PC1 in the PC1 vs. PC3 plot (Figure 2c), and a few pop1 samples were located far away from the other samples along PC3. The above conclusions can be readily observed in the plot with all three axes (Figure 2d). Samples were clearly separated by population, with the exception of a small portion of outliers of pop1 and pop2; the ADMIXTURE analysis for k = 2 divided all Chinese chestnut samples into two different populations.

### 3.5. Phylogenetic Analysis of 185 Chinese Chestnut Landraces

A total of 299,015 SNPs of all chestnut samples were used in the phylogenetic analysis, and samples in the tree were labeled according to the classification in the ADMIXTURE analysis when k = 2 (Figure 1a, Appendix A). The phylogenetic tree shows the evolutionary relationships among the different groups (Figure 3). The evolutionary branches of closely related varieties tended to be clustered within *C. mollissima* (Figure 3). The two populations were separated in the tree; only a few samples were not clustered with their respective populations. According to the clusters in Figure 3, all the individual samples were divided into the north and the south region of Yellow River Basin Region, and were represented in red and blue, respectively. However, some individual samples of the south region (shown in blue) mixed in the north region (shown in red), such as RISF 149, 72, and 24. These individual samples (belonging to the south region which was divided by the geographic position) were divided into the north region by the results of genetic distance, which indicated that different samples had gene interaction. These outlier samples were mainly from Shandong Province and Beijing City (Appendix A). Combining the above results, these outliers were identified for further study and discussion (Appendix A).

## 4. Discussion

In this study, we conducted a genetic analysis of 185 Chinese chestnut landraces to explore the genetic diversity, population structure, and domestication history of Chinese chestnut. Samples were obtained from five geographic regions belonging to 14 different provinces or cities. In a previous study, a high level of genetic diversity and genetic differentiation of 279 chestnut individuals from 10 populations in Shandong province by SSR markers analysis indicated an abundant genetic diversity of Chinese chestnut resources [31]. Genetic diversity and structure analysis of chestnut populations can be a useful strategy for conservation, decision-making, and management planning [31,32]. Moreover, 95 cultivars of Chinese chestnut from ten provinces were analyzed by SSR analysis and showed a high richness in genetic diversity [11]. By estimating the genetic variability of sweet chestnut in southwest Bulgaria, Lusini et al. (2014) [33] showed that a combination of natural events and human impacts affected the genetic diversity and spatial structure. Previous studies highlighted that there was no direct relationship between geographic distribution and genetic diversity, that is, no distinct relationship between the genetic distance and the geographic distance [31]. Our results were consistent with these studies that explain the genetic variation mainly from the intra-population level.

PCA, ADMIXTURE analysis, and phylogenetic analysis were used for genetic analysis of 185 Chinese chestnut landraces. ADMIXTURE analysis of all 299,015 SNPs revealed that k = 2 was the optimal number of groupings. We found that the best k value was 12 based on the CV error. However, Chinese chestnuts were divided into two main types, the north and south regions, according to the geographical, ecological, and climatic conditions, and a variety characteristics when k value was 2. Furthermore, taking the Yellow River as the dividing line between the northern and southern chestnut groups is a relatively correct viewpoint, and when k value was 2 chestnut varieties could be distinguished clearly in this study. Therefore, we divided and analyzed our groups according to the k value 2. We found that a north–south geographic division was consistent with the classification of the samples. Gene flow was detected between the two groups, but some samples from Shandong Province and Beijing City were outliers that were not clustered with their respective groups. PCA and phylogenetic analysis were performed on the SNP data, and the classification of the ADMIXTURE analysis when k = 2 was used to visualize the grouping of populations; the results of these analyses were consistent with those of the ADMIXTURE analysis. Pop1 and pop2 were clearly separated; however, samples from Beijing City and Shandong Province did not fit the north–south geographic division.

Geographical factors have a substantial influence on the characteristics of Chinese chestnut varieties, and the phenotypic diversity of Chinese chestnut varies among geographical groups [11]. Generally, the genetic relationships among chestnut resources are related to their geographical origin, but this is not always the case. Shifts in the economic center of gravity and the migration of people from northern to southern regions have also affected the historical expansion of Chinese chestnut in ancient China. There have been several migrations of people from northern to southern China due to the frequent wars that took place in northern China during the Tang (from the year of 618 to 907) and Song (from the year of 960 to 1279) Dynasties. Chinese chestnut is one of the oldest domesticated fruit trees in China [33], and it is an extremely important source of nuts and grain rich in starch. We speculate that the expansion of the Chinese chestnut benefited from the labor and advanced technology provided by northern immigrants, and the suitable climate and fertile soil in southern China accelerated the spread of chestnut from northern to southern regions. The lack of fit of Chinese chestnut samples from Shandong Province and Beijing City with the north–south geographic division might be explained by other geographical and historical factors. A study of 29 pairs of SSR primers for analyzing the genetic diversity of 26 Chinese chestnuts from different regions in Shandong Province revealed geographic variation in the genetic features. However, a few samples of Chinese chestnut from different regions and even geographically distant samples are closely related genetically; this might stem from similarity in environmental conditions and artificial selection criteria [34]. Shandong Province features both banks of the Yellow River; it is thus an area where chestnuts from both the north and south occur. For example, Chinese chestnuts in Linyi City, Shandong Province are located far from the Yellow River in Shandong Province and belong to the second subpopulation. Chinese chestnuts of Beijing City and Hebei Province belong to the Yanshan Mountain ecotype, and this region has been one of the most important areas for the production of Chinese chestnut since ancient times. On the basis of historical records, the scale of Chinese chestnut planting was very large in the Han dynasty (from 202 BC to 220 AD), and this greatly enriched the genetic diversity of chestnut in the Yanshan region, especially the Yanshan mountain ecotype. Chinese chestnuts in North China gradually expanded westward and southward and penetrated into other ecological populations. Supplies and materials from various places circulate through Beijing, which is the capital of China and the historical political center, and this might explain why the germplasm resources in North China have historically been of great significance to the evolution of chestnut.

In the same way, previous studies have shown that *Castanea* species are greatly affected by geographical factors and human activities. The historical distribution of European chestnut in glacial refugia in the Mediterranean basin has had a major effect on current patterns of genetic diversity [34]. SSR markers have shown that the potential ancestral sources of sweet chestnut in Britain and Ireland are more closely related to lineages from Western Europe rather than Eastern European lineages [35]. The Western European regions of Portugal, Spain, France, Italy, and Romania served as refugia during the Last Glacial Maximum. The geographical distribution of Chinese chestnut is divided by the Yellow River. The physiological characteristics of the recalcitrant seeds (difficult germination and short lifetime) of Chinese chestnut hindered the spread of the seeds over long distances under natural conditions; consequently, seeds can only be effectively spread via the aid of birds or rodents [36,37]. These findings confirmed our speculation that the north–south grouping of chestnut populations was influenced by human activities. Similarly, the population dynamics of sour jujube (*Ziziphus acidojujuba*) and common walnut (*Juglans regia*) were also greatly affected by human behavior [38,39].

Chinese chestnut, which is considered a woody grain, is well known for its short growth cycle, high yield, strong adaptability, and its ability to grow in mountainous areas. With its deep roots and luxuriant leaves, Chinese chestnut can not only increase the income of farmers, but also regulate the climate, fertilize the topsoil, and provide ecosystems services [1]. This suggests that the migration of populations towards the south due to wars might have resulted in the introduction of northern Chinese chestnut populations to the south as a food crop during the Tang and Song Dynasties (from the year of 618 to 1279). This might explain the close genetic relationships between some southern subpopulations and northern subpopulations. The dual function of Beijing City and Hebei Province as political and economic centers has likely enhanced the genetic diversity of chestnut. Chinese chestnut in Shandong Province occurs on both banks of the Yellow River, which might be associated with similar artificial selection criteria [31]. Genetic relationships among chestnut germplasm resources are complicated after long periods of natural selection and artificial breeding [40]. Overall, analyses of greater numbers of chestnut samples and populations might generate additional insights.

## 5. Conclusions

A lack of knowledge of the genetic relationships among chestnut populations, patterns of genetic diversity, and the domestication history of chestnut impedes future chestnut breeding efforts. SNPs, a third-generation molecular marker, were used in this study to explore the population structure of chestnuts in China. Our study showed that the genetic diversity level of the five Chinese chestnut populations from different regions was relatively high in observed heterozygosity. The population in the mid-western regions showed the highest genetic diversity, while the population north of the Yellow River showed the lowest. Molecular variance analysis showed higher variation within the group, which indicated the genetic variation of chestnut mainly from the intra-populations. ADMIXTURE analysis with k = 2 revealed a north–south division of the samples, with the Yellow River as the geographical boundary. The results of the PCA and phylogenetic analysis were consistent with the results of the ADMIXTURE analysis. However, chestnuts from Shandong Province and Beijing City were outliers that were not clustered with their respective groups. Therefore, we speculate that the historical distribution of Chinese chestnut has been shaped by human activities, including several migration events driven by wars. The chestnuts in Shandong Province and Beijing City occur in an area with samples from both northern and southern regions, which is likely related to their geographical and political importance. Ultimately, we aimed to obtain information that could aid the genetic diversity evaluation and conservation strategies of chestnut genetic resources in China.

## Figures and Tables

**Figure 1 plants-11-03524-f001:**
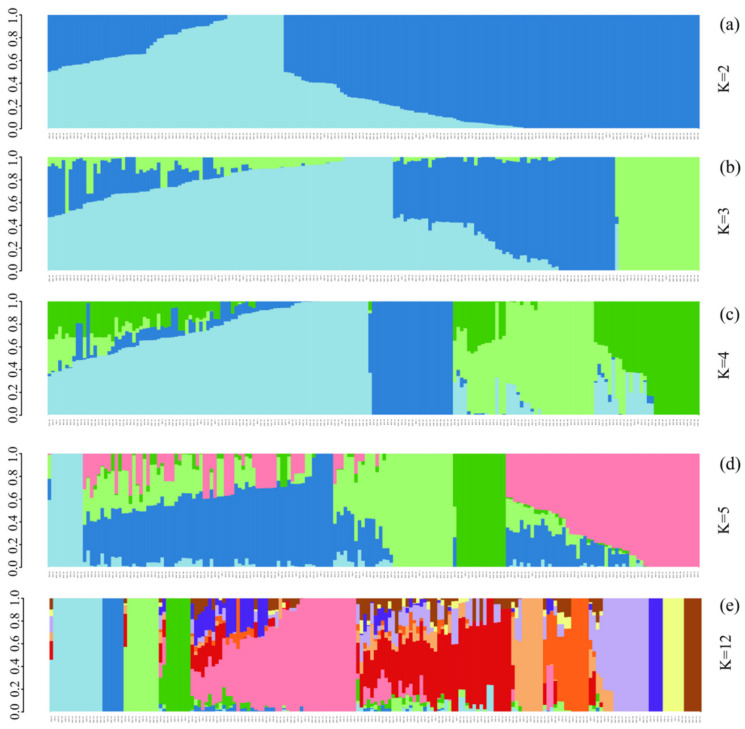
Results of an ADMIXTURE analysis of 185 Chinses chestnut landraces. (**a**) k = 2, (**b**) k = 3, (**c**) k = 4, (**d**) k = 5 and (**e**) k = 12. The K value represents the number of different subgroups, and different colors represent different subgroups.

**Figure 2 plants-11-03524-f002:**
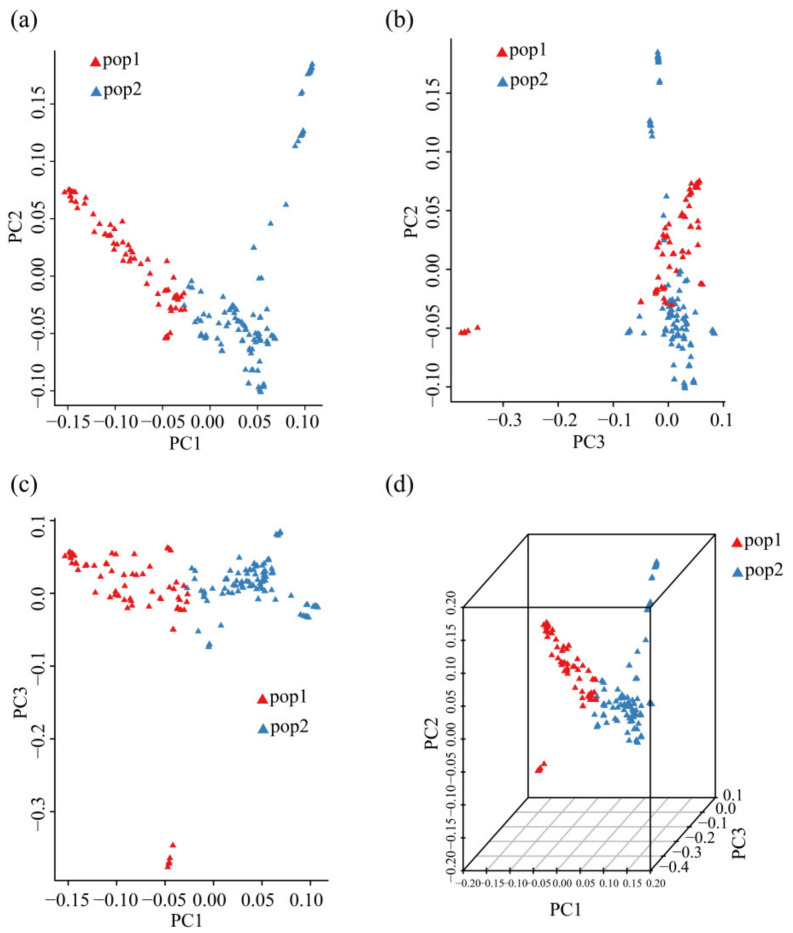
PCA of 185 Chinese chestnut landraces. (**a**) PC1 vs. PC2, (**b**) PC2 vs. PC3, (**c**) PC1 vs. PC3, (**d**) PC1 vs. PC2 vs. PC3.

**Figure 3 plants-11-03524-f003:**
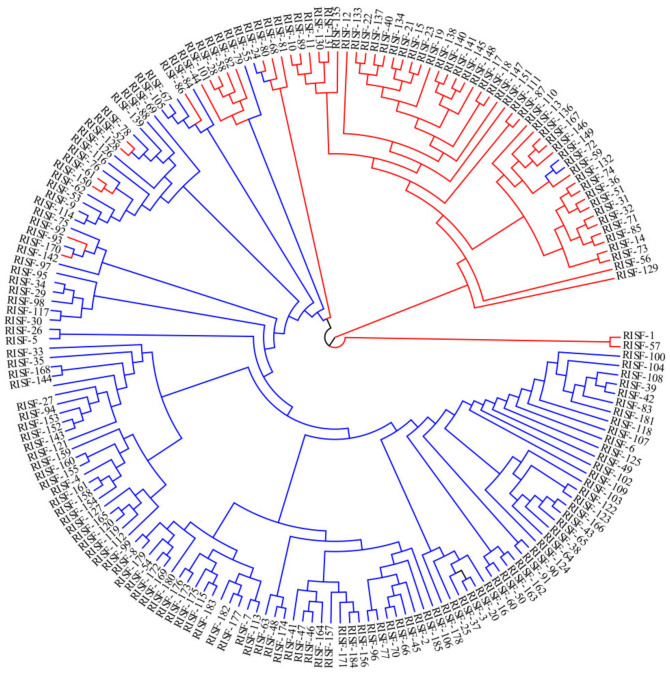
Phylogenetic tree of the 185 Chinese chestnut landraces.

**Table 1 plants-11-03524-t001:** Statistics of sequencing data.

Parameter	Raw Base (bp)	Clean Base (bp)	Effective Rate (%)	Error Rate (%)	Q20 (%)	Q30 (%)	GC Content (%)
Minimum	89,598,816	89,598,816	100	0.02	88.27	81.43	33.47
Maximum	631,849,824	631,849,824	100	0.04	95.97	93.05	39.27
Mean	411,594,134	411,591,572	100	0.03	92.88	89.15	36.58
Total	76,246,654,464	76,246,175,232					

**Table 2 plants-11-03524-t002:** Statistics of sequencing depth and coverage.

Parameter	Clean Reads	Mapped Reads	Mapping Rate (%)	Average Depth (X)	Coverage at Least 1X (%)	Coverage at Least 4X (%)
Minimum	622,214	601,909	91.00	6.51	5.08	0.80
Maximum	4,387,846	4,227,193	98.13	14.75	19.20	6.13
Mean	2,858,275	2,769,234	96.91	9.74	11.80	4.32

**Table 3 plants-11-03524-t003:** SNP statistics and annotation results.

Category		Number of SNPs
Total		299,015
Upstream		4065
Exonic	Stop gain	209
	Stop loss	9
	Non-synonymous	3800
	Synonymous	2241
Intronic		4724
Splicing		47
Downstream		4449
Upstream/Downstream		119
Intergenic		277,994
Transitions (ts)		204,498
Transversions (tv)		94,517
ts/tv		2.163

**Table 4 plants-11-03524-t004:** Genetic diversity level of five populations.

Population	Observed Heterozygosity (*H*o)	Expected Heterozygosity (*H*e)	Gene Flow (*N*m)
North of the Yellow River	0.27963	0.30531	1.7655
Eastern Coastal region	0.30334	0.30612	1.9817
Yangtze River Basin region	0.29995	0.30800	3.2154
South Central region	0.31340	0.30824	2.2518
Midwest region	0.34265	0.33958	1.4372

**Table 5 plants-11-03524-t005:** Analysis of molecular variance of population.

Source of Variation	df	Sum Squares	Variance of Components	Percentage of Variation (%)
Among populations	4	3879.027	8.75943	2.07
Within populations	365	151,380.206	414.74029	97.93
Total	369	155,259.232	423.49972	

## Data Availability

RAW VCF and SRA Database Information can be obtained at request from the corresponding author, all other data are comprised in the manuscript.

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
