# Peer review of "Genetic Diversity and Population Structure of Chinese Chestnut (Castanea mollissima Blume) Cultivars Revealed by GBS Resequencing"

_plants, 2022, doi:10.3390/plants11243524_

Round 1

Reviewer 1 Report

The manuscript by Jiang et al,. is on the application of the GBS method to genotype 185 accessions of Castanea mollissima, identify robust dataset of SNPs to be used for population structure analysis and investigation on genetic variation.

The design of the study is simple but appropriate. The question posed by the authors is clear and the methods that have been used are adequate but often no sufficiently described. The manuscript is generally well written.

I suggest to include:

1)    results from IBS matrix to identify possible duplicated samples (you could use a cut-off >= 0.98 to label samples as duplicated);

2)    results from Treemix (https://bitbucket.org/nygcresearch/treemix/wiki/Home) to estimate the historical relationships (gene flow) among populations.

To respect the principles of open science and data FAIRability, authors MUST submit the reads they got to a public repository (e.g. SRA). In addition, they MUST make the raw vcf or map/ped files available. These are two necessary conditions for me to express a favorable opinion on publication.

Overall, I suggest some compulsory revisions to improve the overall quality of the manuscript.

Line 17: Please, replace “impeded” with “prevented”.

Lines 37-38: In my opinion this statement is not at all true. Please, remove it!

Line 58: After “Chinese chestnut” a reference is missing.

Line 87: The line includes a non-formatted reference.

Line 96: “has made SNPs a particularly important marker”. Plurarl than singular. It sounds scary!

Line 103: the concept has already been expressed before. The line can be eliminated.

Line 130 reference [24]. In my opinion, rather than mentioning GBS experiments in herbaceous plants they should be mentioned for tree plants. See https://doi.org/10.1038/s41598-018-34207-y and https://doi.org/10.1038/s41598-021-02545-z as examples.

Lines 148-149: I suggest to be more adherent to the results of the analyses carried out and to mitigate certain high-rewarding statements. “explore the origin and evolution”????? too pretentious!!

Line 184: Please, remove “clean”before “data”.

Line 186: Please, replache celan reads with high-quality reads

Paragraph 2.4: I remember the authors they are presenting a scientific article and not an educational text. Therefore, please, remove lines 201-211; 216-218; 223-226. instead you should have introduced details about the parameters used by the various tools to ensure the reproducibility of the results. For instance, which k values have been tested with admixture? what threshold value of the membership coefficient (qi) was used to assign an accession to a specific group?

Please, provide settings and parameters for all software used.

Line 239: Please, remove clean before “sequencing data”.

Line 245: the filtering criteria have been introduced in M&M and could be filtered out here.

Lines: 249-252: I repeat: this is not an educational text. Those lines can be eliminated.

Paragraph 3.3:  PLease specify the membership coefficient threshold you did use to assign a sample to a group.  Did you got accessions with admixed ancestry?

Figure 1. What do the colors represent?  I am preety sure you have samples with admixed ancestry. Please, indicate them clearly.

Lines 325-326: This is not clear to me! What does it mean “according to the classification…..”

Lines 341-342: This sentence must be rephrased. 95 accession …were conducted by SSR…… It sounds scary!

Line 352: authors stated that K=2 was the optimal number of groupings. They should show a plot  (supplementary) with the CV error values for all K they tested.

Line 353: “gene flow was…” This claim is pure speculation and it is not based on any data. I suggest you run treemix.

Lines 447-448:  I think the information you got is usless for domestication, cultivation and breeding of chestnut buti t is important for genetic diversity evaluation and conservatuon strategies of genetic resources. Please, rephrase!

Supplementary Table S1: As the authors showed barplot at K=2,3,4,5, I suggest to include in the table also information based on the ramining K values. In addition,  the “admixed” class should be included as well, in column 2.

Table S2: It is not clear to me how these outliers were identified. Please, specify it in the main text

Author Response

Thank you very much for the opportunity that you provided in allowing us to revise our manuscript. We appreciate you and the reviewers very much for the comments on our manuscript entitled “Genetic diversity and population structure of Chinese chestnut (Castanea mollissima Blume) cultivars revealed by GBS resequencing” (Manuscript Number: plants-1963162) in Plants. Those comments are all valuable and very helpful for revising and improving our paper.

We read the reviewer’s comments carefully and tried our best to revise our manuscript according to the comments. Those changes will not influence the content and framework of the paper. Changes and amendments are highlighted in yellow in the revised manuscript. We would like to express our great appreciation to you and reviewers and hope that the correction will meet with approval. We are looking forward to hearing from you soon. If you have any questions, please feel free to contact us at the address below.

Correspondence and telephone call on the paper should be directed to e-mail address: yanpengwang@caf.ac.cn, and the following phone, Tel.:+86-0571-63131182.
Finally, thanks again for your attention to our paper.
Best regards,
Yanpeng Wang

Reviewer 2 Report

The paper describes the application of gbs to investigate genetic diversity and population structure of Chinese chestnut samples covering five larger regions in China. The approach is well suited to this end, and the experimental results are well presented. However, the manuscript could be considerably improved by tighter editing, omission of unsupported conclusions, and some improvement of the language where imprecise or wrong in grammar. Some details are also missing in material and methods and should either be spelled out, or amended by a suitable reference.

L 27  97.93% which indicating that the genetic variation of chestnut mainly from intra-populations

97.93%  indicating that the genetic variation of chestnut is mainly based within populations

L87 (Huang et al. 2022)

is not in the list of references

L100 studies of Chinese chestnut based on SNPs have been conducted [10].

the cited study is using SSR markers! The study LaBonte, Zhao & Woeste (frontiers in plant Science, 2018, vol 9, artice 810) uses SNPs from genome sequencing, and is not cited

L 101 - L144 this part of the introduction should be much shortened, some is repeated from the previos paragraphs, or has little relevance for the argument why this work has been undetaken  

L117 -118 Many researchers have examined population structure and genetic diversity at the level 117 of morphology, isozymes, and DNA [23].

these approaches are around since the early 90's, morphology and isozymes much longer. It makes no sense to cite one specific work (with no bearing on Castanea) instead of one of many excellent reviews / monographs on that topic.

L132 -133 The GBS approach reduces genome complexity through the use of restriction enzymes and DNA-barcoded adapters

this is quite inaccurate. gbs reduces genome complexity by limiting read starts to RE sites. The barcodes enable pooling of sequencing libraries from different sources in one sequencing line. 

L145 GBS was performed on 185 Chinese chestnut landraces from five regions

here and later the authors refer to the origin of their accessions. It would be helpful for the reader to have a overview  map of China which marks out the five regions, possibly with information on collection sites, and distribution range of the species. Are any wild accessions included in this collection? If not it should at least be discussed

L163 DNA extraction

The authors do not specify how DNA was extracted, this needs to be either spelled out, or a reference given 

L167 -168 GBS library was con- structed via various steps:

the authors don't specify how this was done- crucially, which enzymes were used (from L187 I infer one of the enzymes was MseI, but this needs to here where the method is described). Either it was done exactly as published earlier (eg as in Elshire et al, but this is not cited here) , or it needs to be spelled out

L174 When the insert size met expectations

needs to be spelled out, or a reference given

L203-205 Population genetic  diversity analysis can be used to infer the source and degree of differentiation of each  subgroup from phylogenetic analysis, PCA, and population genetic structure analysis.

this is rather vague, I am not sure what is meant- differentiation of a subgroup from its analysis?

L208 A population can be divided into several subspecies

In my view, a subspecies would be a higher order of classification to population

L216 -217 PCA is a dimensionality reduction method that can transform the initial data set into a set of linearly uncorrelated variables. It can compress a large amount of information into two or three major axes of variation for visualization

this is imprecise; PCA does not compress a large amount of information into three but tries to identify major axis of variation, and discards a lot of information in the process, depending on how much of the variability is contained in the first three dimensions (only these can be visualized) 

L226 as all populations share common ancestors

this is a non-seqitur to the preceding sentence

L232 And the 232 total sequencing data volume was

sentence

L238 All samples were not contaminated indicated the library construction was successful

sentence / does contaminated mean "containing adapter sequences" ?

L327  The evolutionary branches of closely related species

all here is within C. mollissima

L333 Discussion

a lot of discussed germplasm movement in view of Chinese history is speculative, this should be much shortened and made more consistent. eg L366 Generally, the genetic relationships among chestnut resources are related to their geographical origin is already contradicted earlier. Much of what follows is only speculation, and in part already repeated from earlier in the manuscript. A later reference to European chestnuts (L398 ff) has no bearing on this article. L413-L422 is also speculative, in part repeated from the introduction. That economic centres like Beijing and Hebei province would have acessions that were brought in from elsewhere is not too surprising

L 448 information that could aid the domestication, breeding, and cultivation of chestnut

domestcation and cultivation has already happened!

Author Response

(The authors gave the same response as above.)

Reviewer 3 Report

Congratulations for your work!

Author Response

Thank you very much.  

Best regards,
Yanpeng Wang

Round 2

Reviewer 1 Report

Dear Authors,
Although the manuscript has improved significantly, I believe it needs additional revisions before being accepted for publication. The main pending issues are related to the the fact that (i) you have not indicated the threshold value of the membership coefficient;

(ii) you seem to have no idea what mixed ancestry individuals are;

(iii) you have not made the data available (raw fastq files and raw vcf  file;)

(iv) based on CV error your best K is 12 and this was not discussed at all;

(v) you have not shown the results of the IBS matrix. you just said there are no duplicates. I would have liked to have at least seen a frequency distribution of the IBS values.

Author Response

Dear Reviewer 1,

Thank you very much for the opportunity that you provided in allowing us to revise our manuscript. We appreciate you and the reviewers very much for the comments on our manuscript entitled “Genetic diversity and population structure of Chinese chestnut (Castanea mollissima Blume) cultivars revealed by GBS resequencing” (Manuscript Number: plants-1963162) in Plants. Those comments are all valuable and very helpful for revising and improving our paper.

We read the reviewer’s comments carefully and tried our best to revise our manuscript according to the comments. Those changes will not influence the content and framework of the paper. Changes and amendments are highlighted in yellow in the revised manuscript. We would like to express our great appreciation to you and reviewers and hope that the correction will meet with approval. We are looking forward to hearing from you soon. If you have any questions, please feel free to contact us at the address below.

Correspondence and telephone call on the paper should be directed to e-mail address: yanpengwang@caf.ac.cn, and the following phone, Tel.:+86-0571-63131182.
Finally, thanks again for your attention to our paper.
Best regards,
Yanpeng Wang

Reviewer 2 Report

While the manuscript has been improved after the first revision, there are still a number of errors and language issues that need to adressed to improve readability prior to publication.

L27 which indicating that the genetic variation of chestnut is mainly based within populations

omit this sentence

L28 All analyses indicated that

omit!

L50 Chinese chestnut, European chestnut, and Japanese chestnut are the main cultivated varieties 

these are species

L52 These varieties also have

species!

L132 Except RISF-72 is a identified hybrid species, others are local cultivated species.

sentence: With exception of the hybrid [of which species?] RISF-72, all other accessions are locally grown C mollissima cultivars.

L135 DNA extraction . 185 135 chestnut young leaves were harvested and immediately frozen in liquid nitrogen and then 136 transformed to -80 ℃. The DNA extraction was used a previous method according to Jiang et al., 137 2017 [12].

[sentence!] DNA extraction. 185 chestnut young leaves were harvested and immediately frozen in liquid nitrogen and then transferred to -80 ℃. The DNA was extracted according to Jiang et al., 2017 [12].

L142 the genome was digested using

genomic DNA was digested

L158 After strict quality control of the sequencing data, high-quality data were obtained.

Remove!

L176 The subspecies based on their geographical distributions or other criteria usually geographically isolated population of organisms.

this is not a sentence

L185 It can compress a large amount of information into 185 two or three major axes of variation for visualization

this is not correct. PCA constructs new axes to represent as much as possible of the variation, but how much of the total variation is reflected in the first 3 dimensions depends on the data set 

L196 used for sequencing analyse. And the total sequencing data volume was

sentence! used for sequencing analysis. The total sequencing data volume was

L198 412.141 M per sample

Mb is correct

L202 All samples were not contaminated adapter sequences indicated the library construction was successful.

[not a complete sentence!] No samples were contaminated by adapter sequences, indicating a successful library construction.

L212 the table need to be tidied up, numbers and category lines are not aligned; abbreviations ts, tv should be spelled out / explaned in the table legend

L217 What’s more, the South central

[not the place for colloquialisms] Furthermore,..

L217 the South central population and the genetic diversity level is close to that of the population of Basin in Yangtze River

[population and diversity are mixed up; is the following meant: ?] the South central population' genetic diversity level is close to that of the population of Basin in Yangtze River

L220 Results of gene flow showed that chestnut in Yangtze River Basin region owned a higher value among them, and the chestnut in Midwest region showed a relative lower value (Table 4).

[language / clarity] Results of gene flow demonstrated higher chestnut genetic diversity in the Yangtze River Basin region, and relatively lower diversity in the Midwest region (Table 4).

L227 carry out statistical tests

carries out 

L233 was 97.93% which indicating that

was 97.93% which indicates that

L234 came from the inside of the five

came from within the five

L273 PC1, PC2, and PC3 explained approximately 17.3%, 9.0%, and 5.5% of the total variance, which captured most of the genetic information of the samples

PC1, PC2 PC3 capture ~1/3 of the genetic information, not most of it!

L295 related species tended to be clustered within

are not all accessions here C.mollissima?

L316 95 chestnut cultivars from ten provinces were conducted by SSR analysis and results also showed the rich genetic diversity

[Sentence is wrong] 95 chestnut cultivars from ten provinces were analyzed by SSR, and also showed the rich genetic diversity

L320 Previous studies have been showed

[grammar] Previous studies have shown

L346 during the Tang and Song Dynasties

[give years for readers less familiar with Chinese history]

L349 benefited from the labor and advanced science and technology provided by

benefited from the labor and advanced technology provided by

[no science in our sense in those days]

L357 distant samples are highly closely related

distant samples are closely related

L366 planting was very large in the Han dynasty

[give years for readers not so familiar with Chinese history]

L381 weak physiological characteristics of the recalcitrant seeds

[meaning difficult to germinate? This needs an explanation.]

L394 during the Tang and Song Dynasties

years?

Author Response

Dear Reviewer 2,

Thank you very much for the opportunity that you provided in allowing us to revise our manuscript. We appreciate you and the reviewers very much for the comments on our manuscript entitled “Genetic diversity and population structure of Chinese chestnut (Castanea mollissima Blume) cultivars revealed by GBS resequencing” (Manuscript Number: plants-1963162) in Plants. Those comments are all valuable and very helpful for revising and improving our paper.

We read the reviewer’s comments carefully and tried our best to revise our manuscript according to the comments. Those changes will not influence the content and framework of the paper. Changes and amendments are highlighted in yellow in the revised manuscript. We would like to express our great appreciation to you and reviewers and hope that the correction will meet with approval. We are looking forward to hearing from you soon. If you have any questions, please feel free to contact us at the address below.

Correspondence and telephone call on the paper should be directed to e-mail address: yanpengwang@caf.ac.cn, and the following phone, Tel.:+86-0571-63131182.
Finally, thanks again for your attention to our paper.
Best regards,
Yanpeng Wang

Round 3

Reviewer 1 Report

Q: you seem to have no idea what mixed ancestry individuals are; 

R: Because almost the samples used in this study are cultivated varieties of Chinese chestnut. Many of them are superior individuals selected from natural populations, which have come from long-term natural selection and artificial domestication. These plants materials have been artificially grafted and propagated for many times. We are so sorry to tell you that it is difficult to find out or identify the source of these materials (their original parents). Therefore, we can not identify these individuals which have mixed ancestors. 

If the fixed membership coefficient threshold is 0.5 you will certainly have individuals for which none of the ancestry exceeds this threshold. These are to be considered admixed individuals and they can be easily identified and highlighted in the population structure bar chart if the allele frequencies are properly sorted.

(iii) you have not made the data available (raw fastq files and raw vcf file;) 

We have contacted the editor (Mr. Ursulescu Vlad Bogdan, Assistant Editor, MDPI Cluj, E-Mail: ursulescu@mdpi.com), and have uploaded the files (chestnut.snp.vcf.gz and chestnut.indel.vcf.gz) to him. 

Well done! but row fastq files need to be submitted to SRA/ENA. That's important for data FAIRability and to ensure the reproducibility of the results.

(iv) based on CV error your best K is 12 and this was not discussed at all; 

We found that the best K value was 12 based on the CV error. However, Chinese chestnut cultivars were mainly divided into two types, the north and the south regions, according to the geographical, ecological and climatic conditions, and variety characteristics when K value was 2. Furthermore, taking the Yellow River as the dividing line between the northern and southern chestnut groups is always well-known as a relatively correct viewpoint, and K value was 2 could be distinguished chestnut varieties clearly in this study. Therefore, we divide and analyze our groups according to the K-value 2 finally, and the results well confirmed this view (Line 325-335). 

Your explanation is clear and it was before. My suggestion is to insert the bar chart also at K=12 and to discuss it in the text. 

(v) you have not shown the results of the IBS matrix. you just said there are no duplicates. I would have liked to have at least seen a frequency distribution of the IBS values. 

We are very sorry that we did not provide relevant results before. According to your requirements, we searched the current calculation result file and provided the IBS file (Fig. S2). No duplicate samples were found. 

The figure alone you provided gives an overview but it is not good. to verify that your statement is correct (no duplicate samples) I need to see the IBS values obtained. Please provide a table to support the figure.

Author Response

Q1: If the fixed membership coefficient threshold is 0.5 you will certainly have individuals for which none of the ancestry exceeds this. These are to be considered admixed individuals and they can be easily identified and highlighted in the population structure bar chart if the allele frequencies are properly sorted.

R1: We are so sorry that we did not understand your requires totally before. It can be easily to identify the admixed individuals from k=2 in according to your suggestion. We thought the admixed individuals were these membership coefficient which didn’t reached to 1.0, and there were  51 and 68 admixed individuals in the north and south regions after the calculation, respectively (Figure 1a).

Q2: you have not made the data available (raw fastq files and raw vcf file;) 

A: Thank you for your useful suggestion. We have uploaded the date to SRA (https://submit.ncbi.nlm.nih.gov/subs/sra/) on 23th/11/2022. But we need some time to wait for the audit by the staff.  

Q3: based on CV error your best K is 12 and this was not discussed at all. Your explanation is clear and it was before. My suggestion is to insert the bar chart also at K=12 and to discuss it in the text. 

A3: Thanks for you useful suggestion. We have added k=12 in Fig.1 and explained the reason why we choose k value = 2 instead of 12 (Line 326-333).

Q4: you have not shown the results of the IBS matrix. you just said there are no duplicates. I would have liked to have at least seen a frequency distribution of the IBS values.  

The figure alone you provided gives an overview but it is not good. to verify that your statement is correct (no duplicate samples) I need to see the IBS values obtained. Please provide a table to support the figure.

A4: Thank you for your suggestion. We are very glad to offer the original date for you. However, we are so sorry that the original data (named plink.mdist) was too large to make a excel. We have sent the file to the editor (Mr. Ursulescu Vlad Bogdan, Assistant Editor, MDPI Cluj, E-Mail: ursulescu@mdpi.com ).  

Round 4

Reviewer 1 Report

All my concerns were satisfactorily addressed. 

It remains to indicate in the text the "data availability statement" with reference to the raw vcf and the SRA database

Author Response

ABSTRACT

  1. L34-35: please rephrase, for example: “The findings of our study showed the complex genetic relationships among Chinese chestnut landraces and the high genetic diversity of these resources.”

Thank you for your useful suggestion. We have corrected it (Line 34-36).

KW

  1. L37: remove”;” after “SNP-based”; change “north-south” to “geographical”

Thank you for your useful suggestion. We have corrected it.

INTRODUCTION

  1. L48: add “European or” before “sweet”

Thank you for your useful suggestion. We have corrected it (Line 47).

  1. L68-71; L72-83: please, merge these two paragraph

Thank you for your useful suggestion. We have merged it.

  1. 5. L69,74: check ref. [10], it is lacking between ref. [9] (L69) and ref. [11] (L74)

We are so sorry to make this mistake. We have corrected it (Line 75).

  1. 6. L88-89: please, explain better what you mean, SNPs have the highest level of resolution among molecular markers, but each SNP is usually a bi-allelic type of marker (see L87), SSRs show many more alleles per locus…

We are so sorry to make this mistake, and we have deleted it to avoid confusion.

  1. 7. L95: change reference [10] to reference [10-12] about use of SSRs in Chinese chestnut, and, please, check for other references in the Ms

We are so sorry to make this mistake, and we have checked it.

  1. 8. L95: here, add “, while more recently research utilized SNPs in European chestnut”. Add the two recent references on the use of SNPs as representative examples for managing the chestnut genetic resources, in the case in point Castanea sativa, both references are of Nunziata et al.:

Thank you for your useful suggestion. We have added them (Line 95).

  1. Nunziata, A.; Ruggieri, V.; Petriccione, M.; De Masi, L. Single Nucleotide Polymorphisms as Practical Molecular Tools to Support European Chestnut Agrobiodiversity Management. Int. J. Mol. Sci. 2020, 21, 4805. https://doi.org/10.3390/ijms21134805
  2. Nunziata, A.; Ferlito, F.; Magri, A.; Ferrara, E.; Petriccione, M. The Hundred Horses Chestnut: a model system for studying mutation rate during clonal propagation in superior plants. Forestry. 2022, 95 (5): 678-685.
  3. L100-103: please, thoroughly check the English language and rephrase

Thank you for your useful suggestion. We have checked it (Line 98-100).

  1. 10. L106: please, check references [22, 23, 24] and change to [20, 22-24], add pepper to the list with ref. [20], and add the plant species cited: olive [23] and Norway spruce [24]

We are so sorry to make this mistake, and we have checked and added them (Line 106).

  1. L117-118: remove the repetition “Chinese chestnut, which was”

Thank you for your useful suggestion. We have removed it (Line 117-118).

M&M

  1. 12. L124-133: a detailed geographic map with regions and Provinces of Chinese chestnut sampling is required

Thank you for your helpful suggestion. We are very sorry we can not to offer the accurate site of these 185 Chinese chestnut landraces because of the strict censorship. We tried our best to description the details in Table S1. And, we have replied to Reviewer 1 and gained his/hers understanding.  

  1. L125-126: add “(C. mollissima)” after the common name, change “chestnuts” to “chestnut”, consider according verb with subject, and add “samples “ after “39”

Thank you for your useful suggestion. We have corrected it (Line 125-129).

  1. 14. L132: if the clone RISF-72 is an inter-specific hybrid, then report the two species of origin, otherwise only “(C. mollissima)” is obvious

Thank you for your useful suggestion. We have added the information (Line 133).

  1. L135: remove “185” (see L125)

Thank you for your useful suggestion. We have removed it.

  1. L136-137: change to “Chestnut young leaves were harvested, immediately frozen in liquid nitrogen, and then transferred to -80°C”

Thank you for your useful suggestion. We have corrected it (Line 136-138).

  1. L137: check font style

We are so sorry to make this mistake, and we have corrected it (Line 138).

L137-140: remove “Three methods were used to process chestnut samples: 1)”, “2)”, and “3)”

Thank you for your useful suggestion. We have removed it.

  1. L142: change “various” to “the following”, add “according to Elshire 2011” before [19]

Thank you for your useful suggestion. We have corrected it (Line 140).

  1. 19. L145: remove “a” before “Qubit”, indicate in brackets the brand and place of origin of each instrument

Thank you for your useful suggestion. We have corrected it.

  1. 20. L147: change “µl” to “µL”, and remove “An” before “Agilent”

Thank you for your useful suggestion. We have corrected it (Line 145).

  1. L150: ref. [26] is not pertinent, it is about SSRs, please fix

Thank you for your useful suggestion. We think we dont need to cite this reference, so we deleted it.

  1. 22. L151: remove “an” before “Illumina”,indicate in brackets the brand and place of origin of each instrument

Thank you for your useful suggestion. We have removed it and added the related information of each instrument (Line 143-150).

  1. 23. L153: please, check the English language, change “for” to “by”

Thank you for your useful suggestion. We have corrected it (Line 151).

  1. 24. L159: check the name of enzyme Mse1 to MseI

We are so sorry to make this mistake, and we have checked and corrected it (Line 158).

  1. 25. L163: explain the acronym BWA

Thank you for your useful suggestion. We have explained it (Line 162).

  1. L167: after ref. [27], [28, 29] are lacking, here and in the reference list, change ref. [30] to [28] and the references below

We are so sorry to make this mistake, and we have corrected it.

  1. L168: avoid repetitions, change “SNPs” to “data” before “SNPs”

Thank you for your useful suggestion. We have changed it.

  1. 28. L171: as above (see L167), here and in the reference list change ref. [28] to [29]

Thank you for your useful suggestion. We have re-order the reference.

  1. L174,176, 177, 188: here and wherever it is in the Ms, change “subspecies” to “subpopulations”

Thank you for your useful suggestion. We have changed them in the MS (Line 173,175,176 and 189).

  1. 30. L177-179: please, check the English language, change to “ADMIXTURE is a program for the maximum likelihood estimation of individual ancestries based on large SNP genotype datasets. Here, it was utilized to analyze the population structure of Chinese chestnut.”

Thank you for your useful suggestion. We have changed it (Line 177-179).

  1. 31. L180: avoid repetition of “file”

Thank you for your useful suggestion. We have changed it to avoid repetition.

R&D

  1. 32. L207: change “was” to “were”

Thank you for your useful suggestion. We have changed it.

  1. 33. L206-207: avoid repetition of “filtering criteria”

Thank you for your useful suggestion. We have changed it to avoid repetition (Line 207).

  1. 34. L240: check for Table 1, should indicate Table S1, please fix

We are so sorry to make this mistake, and we have corrected it (Line 241).

  1. 35. L263: add a close bracket after each letter: a) K=2 etc.

We are so sorry to make this mistake, and we have corrected it (Line 265).

  1. 36. L284: see L263, add a close bracket after each letter: a) PC1 vs. PC2 etc.

Thank you for your useful suggestion. We have corrected it (Line 287).

  1. 37. L282: Figure 2 is very low resolution and not legible, please fix

Thank you for your useful suggestion. We have fixed it.

  1. 38. L301: Figure 3 is very low resolution and not legible, please fix

Thank you for your useful suggestion. We have fixed it.

  1. 39. L288: check for Table 1, should indicate Table S1, please fix (see L240)

We are so sorry to make this mistake, and we have corrected it (Line 291).

DISCUSSION

  1. 40. L306: change “and” to “belonging to”

Thank you for your useful suggestion. We have changed it (Line 310).

  1. 41. L307: insert “In a previous study,” before “A high level…”

Thank you for your useful suggestion. We have added it (Line 311).

  1. 42. L308: insert “Province” after Shandong

Thank you for your useful suggestion. We have added it (Line 312).

  1. 43. L309: add “Chinese” before “chestnut”

Thank you for your useful suggestion. We have added it (Line 309).

  1. 44. L309-310: check the English language, for example change to “Genetic diversity and structure analysis of chestnut populations can be a useful strategy for conservation, decisions-making and management planning [31].”

Thank you for your useful suggestion. We have changed it (Line 314-315).

  1. 45. L311-312: check the English language, for example rephrase to “Moreover, 95 cultivars of Chinese chestnut from ten Provinces were analyzed by SSR analysis and showed a high richness in genetic diversity [12].

Thank you for your useful suggestion. We have rephrased it (Line 316-317).

  1. 46. L313: insert “Lusini et al. (2014) [33]” before “showed”, and remove comma after “that”

Thank you for your useful suggestion. We have corrected it (Line 318-319).

  1. 47. L314: remove “[33,12]”

Thank you for your useful suggestion. We have removed it.

  1. 48. L315: check the English language (showed), change “have showed” to “highlighted”

Thank you for your useful suggestion. We have changed it (Line 320).

  1. 49. L317-318: check the English language, for example rephrase to “Our results were consistent with these studies that explain the genetic variation mainly from the intra-population level.”

Thank you for your useful suggestion. We have rephrased it (Line 322-323).

  1. 50. L319: change “to” to “for”

Thank you for your useful suggestion. We have changed it (Line 324).

  1. 51. L321: remove “most”Line 328

Thank you for your useful suggestion. We have removed it.

  1. 52. L322: remove “cultivars”, use Chinese chestnuts, and remove “mainly”

Thank you for your useful suggestion. We have removed them.

  1. 53. L323: insert “main” after ”two”

Thank you for your useful suggestion. We have inserted them (Line 327).

  1. 54. L326: check the English language, rephrase the concepts more clearly, remove “always well-known”, change “could be distinguished” to “could distinguish”

Thank you for your useful suggestion. We have corrected them (Line 331).

  1. 55. L327: use past tense for divide and analyze

We are so sorry to make this mistake, and we have corrected it (Line 332).

  1. 56. L328: remove “finally”

Thank you for your useful suggestion. We have removed them.

  1. 57. L329: remove “most”

Thank you for your useful suggestion. We have removed it.

  1. 58. L340: the ref. [2] is not pertinent, please fix

We are so sorry to make this mistake, and we have deleted it.

  1. 59. L357: remove repetition of “Shandong Province”

Thank you for your useful suggestion. We have removed them.

  1. 60. L374: add “In the same way,” before “Previous studies”, and remove “of chestnut”

Thank you for your useful suggestion. We have corrected them (Line 377).

  1. 61. L382-384: check the English language, rephrase the concepts more clearly, remove “weak”, explain better: recalcitrant seed are of difficult germination (not easy), check “impede”…

We are so sorry to make this mistake, and we have changed the impede to hindered (Line 386).

  1. 62. L385-388: check the English language, rephrase the concepts more clearly, avoiding repetitions: “affected”, “human activities”, and use past tense

Thank you for your useful suggestion. We have rephrased them (Line 389,391).

  1. 63. L392: add comma before “but”, remove “and”, add: “, and provide ecosystems services”

Thank you for your useful suggestion. We have corrected them (Line 396-397).

  1. 64. L395: fix typo in “form”

We are so sorry to make this mistake. We have corrected it (Line 399).

  1. 65. L396: change “, and” to “.” separating the two sentences;
  2. 66. L399: change “, and” to “.” separating the two sentences

Thank you for your useful suggestion. We have corrected them (Line 400,403).

CONCLUSION

  1. 67. L405-406: remove this sentence

Thank you for your useful suggestion. We have removed it.

  1. 68. L407: add “Chinese chestnut” after “five”

Thank you for your useful suggestion. We have added it (Line 414).

  1. 69. L408-409: check the English language

Thank you for your useful suggestion. We have checked it (Line 415-417).

  1. 70. L413: add “geographical” before “boundary”

Thank you for your useful suggestion. We have added it (Line 420).

  1. 71. L420-423: move these two sentences to the beginning of this section

Thank you for your useful suggestion. We have removed it (Line 410-413)

  1. 72. L423: add “Ultimately,” before “We aimed”

Thank you for your useful suggestion. We have added it (Line 428).

REFERENCES

Q:The citations have some issue. For example in L150, the contents are not consistent with the cited paper. Please, check all the citations thoroughly to ensure that all citations match the contents. Use italics font to indicate the scientific name of plant species (L474, L480…)

A: We are so sorry to make these mistakes. We have fixed it (Line 459,468,473,481,529).

Q: A further suggestion is to add two recent references on the use of SNPs as representative examples for managing the chestnut genetic resources, in the case in point Castanea sativa, both references are of Nunziata et al.:

A: Thanks for your useful suggestion. We have cited these references.